# Prevalence and Clinical Impact of Viral and Bacterial Coinfections in Hospitalized Children and Adolescents Aged under 18 Years with COVID-19 during the Omicron Wave in Russia

**DOI:** 10.3390/v16081180

**Published:** 2024-07-23

**Authors:** Alexander S. Yakovlev, Vladislav V. Afanasev, Svetlana I. Alekseenko, Ilmira K. Belyaletdinova, Ludmila N. Isankina, Irina A. Gryaznova, Anatoly V. Skalny, Liubov I. Kozlovskaya, Aydar A. Ishmukhametov, Galina G. Karganova

**Affiliations:** 1FSASI “Chumakov Federal Scientific Center for Research and Development of Immune-and-Biological Products of RAS” (Institute of Poliomyelitis), 108819 Moscow, Russia; yakovlev_as@yahoo.com (A.S.Y.); lubov_i_k@mail.ru (L.I.K.); ishmukhametov@chumakovs.su (A.A.I.); 2Otolaryngology Department, I.I. Mechnikov North-Western State Medical University, 191015 St. Petersburg, Russia; streetva@gmail.com (V.V.A.); svolga-lor@mail.ru (S.I.A.); 3K.A. Rauhfus Children’s City Multidisciplinary Clinical Center for High Medical Technologies, 191036 St. Petersburg, Russia; isan-ludmila@yandex.ru (L.N.I.); irinagryaznova80@mail.ru (I.A.G.); 4Saint-Petersburg Research Institute of Ear, Throat, Nose and Speech, 190013 St. Petersburg, Russia; 5Research Institute for Systems Biology and Medicine (RISBM), 117246 Moscow, Russia; belyaletdinova_i@mail.ru; 6Department of Medical Elementology, Peoples’ Friendship University of Russia (RUDN University), 117198 Moscow, Russia; skalnyy-av@rudn.ru; 7Center of Bioelementology and Human Ecology, IM Sechenov Moscow State Medical University (Sechenov University), 119146 Moscow, Russia; 8Institute of Translational Medicine and Biotechnology, IM Sechenov Moscow State Medical University (Sechenov University), 119146 Moscow, Russia

**Keywords:** COVID-19, SARS-CoV-2, influenza virus, bacterial pathogens, coinfections, respiratory viruses, pediatrics, adenovirus

## Abstract

The COVID-19 pandemic has altered respiratory infection patterns in pediatric populations. The emergence of the SARS-CoV-2 Omicron variant and relaxation of public health measures have increased the likelihood of coinfections. Previous studies show conflicting results regarding the impact of viral and bacterial coinfections with SARS-CoV-2 on severity of pediatric disease. This study investigated the prevalence and clinical impact of coinfections among children hospitalized with COVID-19 during the Omicron wave. A retrospective analysis was conducted on 574 hospitalized patients aged under 18 years in Russia, from January 2022 to March 2023. Samples from patients were tested for SARS-CoV-2 and other respiratory pathogens using qRT-PCR, bacterial culture tests and mass spectrometry, and ELISA. Approximately one-third of COVID-19 cases had coinfections, with viral and bacterial coinfections occurring at similar rates. Adenovirus and *Staphylococcus aureus* were the most common viral and bacterial coinfections, respectively. Viral coinfections were associated with higher fevers and increased bronchitis, while bacterial coinfections correlated with longer duration of illness and higher pneumonia rates. Non-SARS-CoV-2 respiratory viruses were linked to more severe lower respiratory tract complications than SARS-CoV-2 monoinfection. These findings suggest that during the Omicron wave, seasonal respiratory viruses may have posed a greater threat to children’s health than SARS-CoV-2.

## 1. Introduction

The COVID-19 pandemic, caused by SARS-CoV-2, has led to a global public health crisis and unprecedented epidemic control measures [1]. These restrictive measures, including lockdowns, social distancing, PPE use, and school quarantines, as well as possible viral interference, reduced the incidence of other respiratory viruses such as influenza and respiratory syncytial virus (RSV) [2,3,4,5]. The advent of vaccines and the implementation of effective treatment protocols have led to a relaxation of quarantine measures. This has resulted in the return of many respiratory viruses to their pre-pandemic prevalence rates [4]. Furthermore, the highest incidence rate of the pandemic was observed during the winter of 2021/2022, due to the emergence of the SARS-CoV-2 Omicron variant [6]. Omicron is significantly more transmissible and replicates faster in the respiratory tract than previous SARS-CoV-2 variants [7,8]. Consequently, favorable conditions were established for a higher prevalence of coinfections between SARS-CoV-2 and other respiratory viruses.

Previous studies have demonstrated conflicting findings regarding the impact of viral and bacterial coinfections with SARS-CoV-2 on disease severity in pediatric patients [9,10]. Some studies have suggested a possible association between coinfections and increased complications, while other studies have not found a notable impact. Further data are required to clarify the clinical implications of coinfections with SARS-CoV-2 in the pediatric population. This is particularly important in light of the emergence of new variants such as Omicron.

We aimed to investigate the incidence and clinical characteristics of viral and bacterial coinfections with COVID-19, compared to SARS-CoV-2 monoinfections, in hospitalized children and adolescents during the Omicron wave in St. Petersburg, Russia.

## 2. Materials and Methods

### 2.1. Study Design and Participants

We conducted this retrospective observational study at St. Petersburg Filatov Children’s City Clinical Hospital No. 5, the primary medical facility for admitting children and adolescents with COVID-19 within St. Petersburg. The study covered the period from 9 January 2022 to 10 March 2023, during the spread of the SARS-CoV-2 Omicron variant [11]. During this period, Russia experienced distinct waves dominated by different Omicron subvariants. The initial wave (January–February 2022) was primarily driven by the BA.1 subvariant, followed by a BA.2-dominated wave in spring 2022. Subsequent waves of BA.4 and BA.5 occurred in summer and fall 2022. Towards the end of our study period, in early 2023, the XBB.1.5 sublineage began to increase in prevalence [12]. We randomly selected 574 patients aged under 18 years who were hospitalized due to acute respiratory illness. Our study did not include participants with conditions that significantly increase infection risk, such as cystic fibrosis, immunocompromised states, HIV infection, or those undergoing chemotherapy or immunosuppressive therapy.

We collected nasopharyngeal swabs from all patients within 24 h of admission. RNA/DNA extraction was performed using a RIBO-prep kit, (InterLabService, Moscow, Russia). The specimens were tested for common respiratory viruses by qRT-PCR, including respiratory syncytial virus (RSV), human metapneumovirus (hMPV), adenovirus (AdV), bocavirus (hBoV), rhinovirus (hRV), parainfluenza virus 1–4 (PIV 1–4), coronaviruses (hCoV) 229E/NL63 and HKU1/OC43 (AmpliSens ARVI-screen-Fl kit, InterLabService, Moscow, Russia), influenza A (IAV) and B (IBV) virus (AmpliSens Influenza virus A/B-FL kit, InterLabService, Russia) and SARS-CoV-2 (POLYVIR SARS-CoV-2 kit Lytech, Moscow, Russia). Patients were also tested for various bacterial pathogens using qRT-PCR, including *Streptococcus pneumoniae* (*S. pneumoniae*) and *Haemophilus influenzae* (*H. influenzae*) (AmpliSens Pneumo-Quantum-Fl kit, InterLabService, Russia), *Mycoplasma pneumoniae* (*M. pneumoniae*) and *Chlamydophila pneumoniae* (*C. pneumoniae*) (AmpliSens Mycoplasma pneumoniae/Chlamydophila pneumoniae-Fl kit, InterLabService, Moscow, Russia), *Bordetella pertussis, Bordetella parapertussis*, and *Bordetella bronchiseptica* (AmpliSens Bordetella multi-Fl kit, InterLabService, Moscow, Russia), and *Streptococcus pyogenes* (*S. pyogenes*) (AmpliSens Streptococcus pyogenes-screen/monitor-Fl kit, InterLabService, Moscow, Russia). In addition to qRT-PCR, bacterial culture and mass spectrometry were used to identify a range of bacterial pathogens, including *Staphylococcus aureus* (*S. aureus*), *Streptococcus viridans* (*S. viridans*), *Klebsiella pneumoniae* (*K. pneumoniae*), *H. influenza, C. pneumoniae*, *M. pneumoniae*, etc. For some patients, serum samples were further analyzed for the presence of IgM and IgA antibodies against *M. pneumoniae* and *C. pneumoniae* using enzyme-linked immunosorbent assay (ELISA) kits (Mycoplasma pneumoniae-IgM-IFA-BEST, Mycoplasma pneumoniae-IgA-IFA-BEST, Chlamydophila pneumoniae-IgM-IFA-BEST, and Chlamydophila pneumoniae-IgA-IFA-BEST; Vector Best, Novosibirsk, Russia).

In this study, we use the term ‘coinfection’ to describe the simultaneous detection of SARS-CoV-2 and other pathogens. However, we acknowledge that, particularly in the case of bacterial pathogens, the presence of genetic material or culturable organisms does not always indicate active infection. In some instances, especially for common colonizers of the respiratory tract, our detection may represent co-detection rather than coinfection.

### 2.2. Statistical Analysis

Clinical data were extracted from medical records and analyzed. Statistical analysis was performed using GraphPad Prism 9.0.0 software. Patients were categorized based on positive test results for SARS-CoV-2, other viruses, or bacterial pathogens. Descriptive statistics were employed for demographic characteristics. Continuous data were presented as medians and interquartile ranges, while categorical data were presented as numbers and percentages. Comparisons were made using Mann–Whitney U test or one-way ANOVA for continuous variables, and the chi-square test or Fisher’s exact test for categorical variables. A *p* value < 0.05 was considered statistically significant.

## 3. Results

### 3.1. Pathogen Distribution

A total of 574 pediatric patients, aged 17 days to 18 years, were hospitalized, with symptoms of respiratory disease. Among these, 320 (55.7%) were male, and 254 (44.3%) were female. The median age was 4 IQR [2,11] years.

The number of hospitalizations reached its highest point during the winter–spring and fall–winter periods. A total of 135 individuals (23.5%) were hospitalized from January to March 2022, while 250 individuals (43.6%) were hospitalized from November to February 2023.

Among the 381 cases of COVID-19, 257 (67.5%) were SARS-CoV-2 monoinfections, 48 (12.6%) had coinfections with another respiratory virus, 60 (15.7%) had bacterial coinfections, and 16 (4.1%) had both viral and bacterial coinfections with SARS-CoV-2 (Figure 1). Furthermore, 92 cases of other respiratory viruses were identified, including 74 (80.4%) monoinfections and 18 (19.6%) bacterial coinfections. No viruses were identified in the remaining 101 patients.

Viral coinfections were present in 16.8% of COVID-19 cases. The most common were SARS-CoV-2 and AdV (6.56%), followed by hRV (2.36%) and PIV 3 (2.36%) (Table 1). Among patients who tested negative for SARS-CoV-2, hRV had the highest incidence rate (11.92%), followed by IBV (10.88%), RSV (8.29%), and AdV (8.29%).

Viral coinfections with SARS-CoV-2 were most prevalent in patients aged 1 to 4 years, accounting for 64.04%, *p* < 0.05 (Table 2). The lowest number of cases were observed in children under one year old, with a prevalence of 4.69%.

Bacterial coinfections were reported in 19.9% of patients diagnosed with COVID-19. *S. aureus* accounted for over half of the cases, with a prevalence of 12.07%, followed by *S. viridans* (3.67%) and *S. pneumoniae* (2.36%) (Table 3). Age-wise, bacterial coinfections were most common in children aged 1–4 years (40.79%), whereas the incidence rate for children under 1 year was relatively low (7.89%) (Table 4).

In 4.2% of cases, there was a coinfection of SARS-CoV-2 with both viral and bacterial pathogens. *S. aureus* represented the bacterial infection in 69% of cases, while AdV (31%), IAV (19%), and hBoV (19%) represented the viral infection. This type of coinfections was more common in children aged 1–4 years (39%).

### 3.2. The Impact of Coinfections on Clinical Manifestations and Disease Course

Patients were divided into five groups based on the identified infectious agents, as shown in Table 5. This approach allowed us to study the impact of coinfections on clinical manifestations and disease course, and to compare the burden of disease between children with COVID-19 and those with other respiratory viruses.

#### Characterization of the Study Groups

Group 1: Patients with SARS-CoV-2 infection only (*n* = 257). The median age was 5.0 [1.0; 13.0] years, with 52% reported as male.

Group 2: Patients with SARS-CoV-2 and other respiratory viruses (*n* = 48).

The median age was 3.0 IQR [1.0; 4.5] years, with 65% reported as male. Among the seasonal respiratory virus coinfections with SARS-CoV-2, the most frequently detected were AdV (41.7%), PIV 3 (16.7%) and hRV (14.6%).

Group 3: Patients with SARS-CoV-2 and bacterial coinfection (*n* = 60).

The median age was 5.0 IQR [2.0; 12.0] years, with 53% reported as male. Bacterial coinfections most commonly included *S. aureus* (58.3%), *S. viridans* (18.3%) and *S. pneumoniae* (11.7%).

Group 4: Patients with other respiratory viruses without SARS-CoV-2 or bacterial coinfections (*n* = 74).

The median age was 3.0 IQR [2.0; 7.0] years, with 59% reported as male. The respiratory viruses were predominantly represented by hRV (25.7%), IBV (20.3%) and AdV (17.6%).

Group 5: Patients with SARS-CoV-2, other respiratory viruses and bacterial coinfection (*n* = 16).

The median age was 3.0 IQR [2.0; 9.75] years, with 63% reported as male. Among the respiratory viruses the most frequently detected were AdV (31.3%), hBoV (18.8%) and IAV (18.8%), while bacterial infections were predominantly represented by *S. aureus* (68.8%), *S. viridans* (12.5%) and *S. pneumoniae* (12.5%). The most common combinations were SARS-CoV-2 + AdV + *S. aureus* (25%) and SARS-CoV-2 + IAV + *S. aureus* (18.8%).

### 3.3. Comparison of Clinical Characteristics between Study Groups

#### 3.3.1. SARS-CoV-2 Monoinfection vs. SARS-CoV-2 with Other Respiratory Viruses

In the SARS-CoV-2 monoinfection group, the children were older on average compared to the group with respiratory virus coinfection: 5.0 IQR [1.0; 13.0] vs. 3.0 IQR [1.0; 4.5] years, *p* < 0.01. The median maximum temperature was significantly higher in the coinfection group: 39.35 IQR [38.85; 39.60] vs. 38.90 IQR [37.80; 39.40] °C, *p* < 0.05. Bronchitis was diagnosed significantly more frequently in the coinfection group: 16% vs. 3%, *p* < 0.01. Other clinical manifestations, including symptoms, duration of illness, and blood markers, did not differ significantly.

#### 3.3.2. SARS-CoV-2 Monoinfection vs. Bacterial Coinfections

These groups had similar ages, symptoms and blood markers overall. However, the duration of illness was significantly longer in the bacterial coinfections group: 12.00 IQR [9.25; 14.00] vs. 9.00 IQR [7.00; 13.00] days, *p* < 0.01, which was the highest value among all examined groups. Pneumonia was also diagnosed much more frequently in the bacterial coinfections group: 15% vs. 3%, *p* < 0.01.

#### 3.3.3. SARS-CoV-2 vs. Other Respiratory Viruses as Monoinfections

The greatest differences were observed between these groups. In the SARS-CoV-2 group, the patients were older: 5.0 IQR [1.0; 13.0] vs. 3.0 IQR [2.0; 7.0] years, *p* < 0.01. Symptoms including runny nose (68% vs. 51%), cough (78% vs. 48%) and shortness of breath (28% vs. 6%) were more characteristic of the other respiratory viruses group (all *p* < 0.0001). In contrast, sore throat (16% vs. 5%) and headache (14% vs. 3%) were more common in COVID-19 patients, *p* < 0.05.

Importantly, lower respiratory tract complications were noted significantly more often in patients with other respiratory viruses compared to SARS-CoV-2 monoinfection, including pneumonia (12% vs. 3%, *p* < 0.05), bronchitis (24% vs. 3%, *p* < 0.0001), and respiratory failure (18% vs. 2%, *p* < 0.0001). Significant differences in blood counts were also found, with lower white blood cell counts in the children with COVID-19 group: 7.00 IQR [5.10; 9.80] vs. 9.75 IQR [6.18; 13.95] ×10^9^/L, *p* < 0.0001. Leukocytosis was more prevalent among patients with other respiratory viruses (34% vs. 16%), *p* < 0.01.

#### 3.3.4. SARS-CoV-2 vs. SARS-CoV-2 + Other Respiratory Viruses and Bacterial Coinfection

Sore throat was more frequent in cases of SARS-CoV-2 monoinfections (16% vs. 6%, *p* < 0.05); headache appeared to be exclusive to monoinfections (14% vs. 0%, *p* < 0.05). Conversely, abdominal pain was significantly more common in coinfections (19% vs. 5%, *p* < 0.05), and pneumonia was diagnosed significantly more frequently in the coinfection group (19% vs. 3%, *p* < 0.05).

### 3.4. Expected versus Observed Co-Detections with SARS-CoV-2

Analysis of co-detection rates between SARS-CoV-2 and other respiratory viruses during co-circulation revealed significantly lower than expected rates for hRV (observed 2 vs. expected 10, *p* < 0.05) and IBV (observed 0 vs. expected 8, *p* < 0.05), suggesting potential viral interference (Table 6).

The expected number of coinfections was defined as the product of the prevalence of virus 1 and the prevalence of virus 2, multiplied by the total sample size. We then used Fisher’s exact test (significance threshold, *p* < 0.05) to assess whether there was a significant difference between the actual and expected number of coinfections.

## 4. Discussion

This retrospective study investigated the prevalence and clinical impact of viral and bacterial coinfections among pediatric patients hospitalized with COVID-19 during the Omicron variant wave in St. Petersburg, Russia. Our findings highlight that while SARS-CoV-2 remains a concern, non-SARS-CoV-2 respiratory viruses posed a significant threat to children’s health during this period.

Overall, 16.8% of COVID-19 cases had viral coinfections, while 19.9% had bacterial coinfections. The majority of coinfections occurred in young children aged 1–4 years. These results align with other pediatric research conducted during the Omicron variant emergence, which noted coinfection rates ranging from 11% to 20% [13,14,15,16].

Significant differences were noted when comparing clinical features between monoinfections and coinfections. Children with SARS-CoV-2 and viral coinfections were younger and had higher fever than those with SARS-CoV-2 monoinfection. Although the incidence of bronchitis was higher in the coinfection group, the duration of illness was shorter. These data do not support increased illness severity in children with viral coinfections, despite some previous studies suggesting otherwise [13,17,18].

Some studies have shown more severe illness in SARS-CoV-2/influenza coinfections in adults [19,20]. In our study, we observed cases of SARS-CoV-2/IAV coinfection without complications, while IBV occurred only as monoinfection. This suggests that, unlike in adults, SARS-CoV-2/IAV coinfection may not be an aggravating factor in children.

However, children with bacterial coinfections appear to be at greater risk for severe disease, with higher pneumonia incidence and longer duration of illness. These findings support the significance of secondary bacterial infections in worsening COVID-19, as reported previously [15,16].

From the beginning of the COVID-19 pandemic, SARS-CoV-2 was considered the respiratory virus with the highest risk of severe outcomes in children [21]. However, our study suggests that this may no longer be the case following the emergence of the Omicron variant. A comparison between monoinfections of SARS-CoV-2 and other respiratory viruses demonstrated that the group with non-SARS-CoV-2 respiratory viruses appeared to experience more severe illness overall. Patients with respiratory viruses had significantly higher rates of lower respiratory tract complications compared to those with SARS-CoV-2 monoinfection, including pneumonia, bronchitis, and respiratory failure. This underscores the ongoing threat of common seasonal respiratory viruses, often eclipsed by the pandemic. Notably, these non-SARS-CoV-2 viruses also triggered a more robust cellular immune response, as evidenced by elevated white blood cell counts and leukocytosis.

Previous studies have shown milder illness with the Omicron variant compared to prior SARS-CoV-2 variants, possibly due to reduced replicative capability in human lung cells but more efficient bronchial replication [7,8,13]. Consequently, Omicron has a higher affinity for the upper respiratory tract than for the lungs. This characteristic elevates its transmissibility but leads to a more positive prognosis. Our findings indicate a low frequency of lower respiratory tract complications among children with COVID-19, providing further support for this assumption.

Notably, the study identified significantly lower than expected co-detection rates of SARS-CoV-2 with hRV and IBV, indicating a potential viral interference that warrants further investigation. Previous studies have suggested that viral interference may occur through various mechanisms, such as competition for host receptors and resources or induction of innate immune responses that inhibit the replication of other viruses [22,23,24,25]. Understanding these interactions could provide valuable insights into viral dynamics and guide the development of novel preventive strategies.

Our study has several limitations that should be considered when interpreting the results. The retrospective design, single-center study and limited sample size may limit the generalizability of our results. Future prospective multicenter studies with a larger sample size could help to confirm and extend our observations.

In conclusion, our findings indicate that during the Omicron wave, non-SARS-CoV-2 respiratory viruses were a substantial cause of severe lower respiratory tract complications in children, challenging the perception of SARS-CoV-2 as the predominant threat. The data suggest a need for continued vigilance and preventive measures against a broad range of respiratory pathogens to protect children’s health effectively.

## Figures and Tables

**Figure 1 viruses-16-01180-f001:**
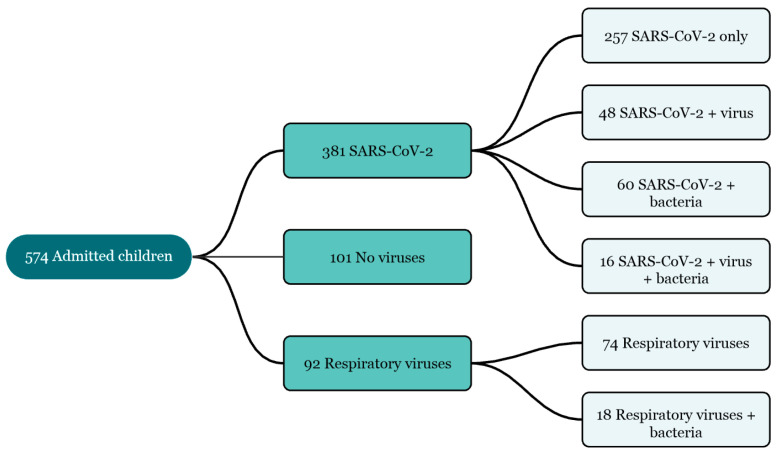
Distribution of patients by pathogen groups.

**Table 1 viruses-16-01180-t001:** Summary of viral coinfection with SARS-CoV-2.

Pathogen	*n*	SARS-CoV-2 Positive *n* = 381	SARS-CoV-2 Negative *n* = 193
AdV	41	25 (6.56%)	16 (8.29%)
hRV	32	9 (2.36%)	23 (11.92%)
PIV 3	9	9 (2.36%)	0 (0.00%)
hBoV	20	7 (1.84%)	13 (6.74%)
IAV	14	6 (1.57%)	8 (4.15%)
RSV	22	6 (1.57%)	16 (8.29%)
hCoV HKU-1/OC 43	12	4 (1.05%)	8 (4.15%)
hCoV NL63/229E	2	2 (0.52%)	0 (0.00%)
hMPV	13	1 (0.26%)	12 (6.22%)
IBV	21	0 (0.00%)	21 (10.88%)
PIV 2	1	0 (0.00%)	1 (0.51%)

**Table 2 viruses-16-01180-t002:** Age distribution of patients with viral coinfections with SARS-CoV-2.

Age, Years	All	Viral Coinfection		*p* Value
		Yes	No	
*n*	381	64	317	<0.0001
Under 1	34 (8.92%)	3 (4.69%)	31 (9.78%)	<0.0001
1–4	160 (41.99%)	41 (64.06%)	119 (37.54%)	<0.0001
5–11	89 (23.36%)	15 (23.44%)	74 (23.34%)	<0.0001
12–17	98 (25.72%)	5 (7.81%)	93 (29.34%)	<0.0001

**Table 3 viruses-16-01180-t003:** Summary of bacterial coinfection with SARS-CoV-2.

Pathogen	*n*	SARS-CoV-2 Positive *n* = 381	SARS-CoV-2 Negative *n* = 193
*S. aureus*	64	46 (12.07%)	18 (9.33%)
*S. viridans*	18	14 (3.67%)	4 (2.07%)
*S. pneumoniae*	13	9 (2.36%)	4 (2.07%)
*Acinetobacter* sp.	4	4 (1.05%)	0 (0.00%)
*H. influenzae*	5	3 (0.79%)	2 (2.07%)
*S. pyogenes*	4	3 (0.79%)	1 (0.51%)
*C. Pneumoniae*	4	2 (0.52%)	2 (2.07%)
*C. albicans*	2	2 (0.52%)	0 (0.00%)
*K. pneumoniae*	3	1 (0.26%)	2 (2.07%)
*Enterobacter* sp.	2	1 (0.26%)	1 (0.51%)
*M. pneumoniae*	2	1 (0.26%)	1 (0.51%)
*E. faecalis*	1	1 (0.26%)	0 (0.00%)
*S. pasteuri*	1	1 (0.26%)	0 (0.00%)
*S. mitis/S.oralis*	1	1 (0.26%)	0 (0.00%)
*P. Aeruginosa*	2	0 (0.00%)	2 (2.07%)

**Table 4 viruses-16-01180-t004:** Age distribution of patients with bacterial coinfections with SARS-CoV-2.

Age, years	All	Bacterial Coinfection		*p* Value
		Yes	No	
*n*	381	76	305	<0.0001
Under 1	34 (8.92%)	6 (7.89%)	28 (9.18%)	<0.0001
1–4	160 (41.99%)	31 (40.79%)	129 (42.30%)	<0.0001
5–11	89 (23.36%)	18 (23.68%)	71 (23.28%)	<0.0001
12–17	98 (25.72%)	21 (27.63%)	77 (25.25%)	<0.0001

**Table 5 viruses-16-01180-t005:** Comparison of demographic profiles, clinical features and symptoms between groups.

	1. SARS-CoV-2	2. SARS-CoV-2 + Virus	3. SARS-CoV-2 + Bacteria	4. Respiratory Viruses	5. SARS-CoV-2 + Virus + Bacteria
*n*	257	48	60	74	16
Study participants, Median IQR or %					
Age, years	5 [1.0; 13.0]	3 [1.0; 4.5] *	5 [2.0; 12.0]	3 [2.0; 7.0] *	3 [2.0; 9.750]
Sex, Male	0.52	0.65	0.53	0.59	0.63
Symptoms					
Fever	0.87	0.92	0.93	0.88	0.81
Runny nose	0.51	0.57	0.53	0.68 *	0.44
Cough	0.48	0.41	0.55	0.78 *	0.63
Sore throat	0.16 *	0.22	0.25	0.05	0.06
Headache	0.14 *	0.08	0.18	0.03	0.0
Hoarseness	0.09	0.03	0.08	0.15	0.0
Weakness	0.09	0.14	0.17	0.12	0.06
Vomiting	0.08	0.08	0.07	0.04	0.13
Shortness of breath	0.06	0.05	0.1	0.28 *	0
Abdominal pain	0.05	0.14	0.02	0.09	0.19 *
Temperature, °C	38.90 [37.80; 39.40]	39.35 [38.85; 39.60] *	38.80 [38.03; 39.20]	39.10 [38.00; 39.70]	38.85 [38.00; 39.35]
Duration of illness, days	9 [7.00; 13.00]	7 [7.00; 10.50]	12 [9.25; 14.00] *	10 [7.00; 13.00]	10.5 [8.0; 15.25]
Duration of fever, days	3 [1.00; 4.00]	3 [2.00; 4.00]	3 [2.00; 5.00]	3 [2.00; 4.00]	4 [3.0; 5.0]
Respiratory failure	0.02	0.05	0	0.18 *	0.06
Bronchitis	0.03	0.16 *	0.03	0.24 *	0
Pneumonia	0.03	0.03	0.15 *	0.12 *	0.19 *
Complete blood count					
WBC, × 10^9^/L	7 [5.100; 9.800]	9 [7.650; 12.95]	7.9 [5.80; 12.10]	9.75 [6.18; 13.95] *	7.75 [6.43; 11.55]
WBC ↑	0.16	0.3	0.27	0.34 *	0.19
WBC ↓	0.15	0.03	0.17	0.09	0.13
Hemoglobin, g/L	127.0 [120.0; 134.0]	122.0 [115.5; 127.0]	129.0 [121.0; 139.0]	123.5 [115.8; 133.0]	132.5 [124.0; 137.5]
Hemoglobin ↓	0.08	0.11	0.1	0.05	0.06
Platelets, × 10^9^/L	271.0 [221.0; 340.5]	286.0 [218.5; 337.0]	273.5 [210.3; 360.0]	297.5 [233.0; 361.5]	282.0 [250.5; 364.3]
Platelets ↑	0.17	0.11	0.12	0.16	0.13
ESR	15 [10.00; 22.00]	17 [10.00; 25.50]	13 [8.25; 22.75]	16 [11.00; 22.50]	16.0 [9.25; 21.50]
ESR ↑	0.44	0.57	0.42	0.53	0.5
CRP ↑	0.09	0.19	0.1	0.05	0.13

* Differences are statistically significant *p* < 0.05.

**Table 6 viruses-16-01180-t006:** Expected versus observed co-detections with SARS-CoV-2.

Virus	Expected Co-Detections	Observed Co-Detections	Chi-Square Test, *p* Value
IBV	8	0	0.0075 *
hMPV	5	0	0.0615
hRV	10	2	0.0367 *
RSV	8	4	0.3829
hBoV	6	2	0.2858
hCoV HKU-1/OC 43	4	2	0.6860
IAV	4	2	0.6860
AdV	8	7	>0.9999

* Differences are statistically significant (Fisher’s exact test, *p* < 0.05).

## Data Availability

The data presented in this study are available on request from the corresponding author due to privacy.

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
