# Peer review of "Prevalence and Clinical Impact of Viral and Bacterial Coinfections in Hospitalized Children and Adolescents Aged under 18 Years with COVID-19 during the Omicron Wave in Russia"

_viruses, 2024, doi:10.3390/v16081180_

Round 1

Reviewer 1 Report

Comments and Suggestions for Authors

How was the diagnosis of bacterial infection established in SARS-CoV-2 and bacterial coinfections? The identification of a bacteria in the nasopharynx using qRT-PCR does not indicate that this bacteria is causing an infection, since its presence as a colonizer is common. Instead of SARS-CoV-2 co-infection with bacteria, this group should be called co-detection of SARS-CoV-2 with bacteria, pointing out the probability that in most cases the bacteria were a mere colonizer. This fact, however, is also important since other studies have shown greater severity of Mycooplasma pneumoniae infections when there is co-detection of the Haemophilus influenza bacteria.

How was bronchitis defined? Does it include bronchiolitis, tracheobronchitis and recurrent wheezing episodes? Also asthma attacks? The term bronchitis is too nonspecific and, consequently, little used in pediatric literature.

Author Response

Thank you for your valuable feedback and suggestions. We appreciate your thorough review of our manuscript and the opportunity to address your concerns. Please find our responses below:

Comments 1: How was the diagnosis of bacterial infection established in SARS-CoV-2 and bacterial coinfections? The identification of a bacteria in the nasopharynx using qRT-PCR does not indicate that this bacteria is causing an infection, since its presence as a colonizer is common. Instead of SARS-CoV-2 co-infection with bacteria, this group should be called co-detection of SARS-CoV-2 with bacteria, pointing out the probability that in most cases the bacteria were a mere colonizer. This fact, however, is also important since other studies have shown greater severity of Mycooplasma pneumoniae infections when there is co-detection of the Haemophilus influenza bacteria.

Response 1: We appreciate your attention to the important distinction between co-detection and coinfection. We acknowledge that the presence of bacterial DNA in nasopharyngeal samples, as detected by qRT-PCR, does not necessarily indicate an active infection, as these bacteria can indeed be colonizers. The term "co-detection" would indeed be more precise in describing the mere presence of multiple pathogens.

However, we chose to use the term "coinfection" in our study for several reasons:

  1. "Coinfection" is widely used in similar contexts in scientific publications, allowing our study to be more readily comparable and searchable within the existing body of research.
  2. While we cannot definitively prove that each detected bacterium caused active infection, numerous studies have demonstrated that the presence of bacterial pathogens, even as colonizers, can lead to more severe outcomes in viral respiratory infections.
  3. Our study employed not only qRT-PCR, bacterial culture and mass spectrometry, but also in some cases, serological tests. This multi-faceted approach provides a stronger basis for using the term "coinfection" compared to studies relying solely on PCR detection.
  4. Using "coinfection" maintains consistency throughout the paper, enhancing readability for a broad scientific audience.

However, we recognize the importance of acknowledging the limitation you've pointed out. To address this, we propose adding a statement in our methods section clarifying that while we use the term "coinfection," we acknowledge that in some cases, particularly for bacterial pathogens, this may represent co-detection of colonizing organisms rather than active coinfection.

Comments 2: How was bronchitis defined? Does it include bronchiolitis, tracheobronchitis and recurrent wheezing episodes? Also asthma attacks? The term bronchitis is too nonspecific and, consequently, little used in pediatric literature.

Response 2: In our study, bronchitis was defined based on clinical presentation: symptoms of acute respiratory infection accompanied by diffuse dry and moist rales on lung auscultation, with the absence of focal and infiltrative changes on chest radiography or CT scans.

Our classification of bronchitis encompassed bronchiolitis and tracheobronchitis. However, it did not include asthma attacks or recurrent wheezing episodes unrelated to acute infection.

We acknowledge that the term "bronchitis" may be considered broad in pediatric literature. In our study, we used this term to capture conditions involving diffuse infectious involvement of the lower respiratory tract. This usage aligns with the ICD-10 coding system (code J20) which is standard in the Russian Federation's healthcare system.

Due to the retrospective nature of our analysis of medical records, we were unable to influence the specificity of the diagnoses made by the treating physicians. We acknowledge that for younger children, especially those under two years of age, the term 'bronchiolitis' is often considered more appropriate in pediatric literature. However, in our healthcare system, physicians frequently use the broader term 'bronchitis' for lower respiratory tract infections across various age groups.

Reviewer 2 Report

Comments and Suggestions for Authors

This manuscript delineates the characteristics of viral and bacterial coinfections in hospitalised paediatric patients in St. Petersburg, Russia. It is well-written, well-structured, and informative. The findings cover the most common paediatric respiratory infections and their outcomes in detail.

Major Concerns.

1. Did this study include participants who have a high risk of infection, such as those with cystic fibrosis, immunocompromised conditions, PLWH, or those undergoing chemotherapy/immunosuppressants?
If so, I suggest excluding these participants and creating a new section in the body text and table (which may be added as supplementary materials) to show the outcomes of this group.

Comments

1. English Style: This manuscript contains both UK (e.g., haemoglobin) and US (e.g., hospitalized, pediatric) styles. I suggest choosing only one and making it consistent throughout the manuscript.

2. Influenza C: Did the test for influenza A (IAV) and B (IBV) viruses (AmpliSens Influenza virus A/B-FL kit) cover or cross-react with Influenza C?
Although this virus is rare and less symptomatic than influenza A and B, it would be interesting if your test included it.

https://www.sciencedirect.com/science/article/abs/pii/S0163445314000863

https://wwwnc.cdc.gov/eid/article/12/10/05-1170_article

3. Vaccination Data: This study collects data from hospitalised paediatric patients. It would be valuable to show the vaccination data (e.g., COVID-19, influenza, Hib). This data may reveal the pattern of infection and/or coinfections in these participants.

4. Table 5: Clinical Manifestations: Did this study collect data on anosmia and ageusia?
These two symptoms are prominent and seem specific to COVID-19 even in vaccinated individuals.

5. Table 5: Age Groups: I suggest subgrouping by age following Tables 2 and 4 to make it more informative.
Moreover, each age group could have different risk behaviours and lifestyles, such as infants-toddlers, preschoolers, and school-age children.

6. Lines 68-71: I suggest adding a statement to specify the prominent waves of Omicron subvariants (e.g., BA.2.75, BA.5, etc.) during the study period. Omicron has many subvariants, and each may have different characteristic patterns.

Author Response

Thank you for your valuable feedback and suggestions. We appreciate your thorough review of our manuscript and the opportunity to address your concerns. Please find our responses below:

Comments 1: Did this study include participants who have a high risk of infection, such as those with cystic fibrosis, immunocompromised conditions, PLWH, or those undergoing chemotherapy/immunosuppressants? If so, I suggest excluding these participants and creating a new section in the body text and table (which may be added as supplementary materials) to show the outcomes of this group.

Response 1: Our study did not include participants with conditions that significantly increase infection risk, such as cystic fibrosis, immunocompromised states, HIV infection, or those undergoing chemotherapy/immunosuppressive therapy. We will incorporate this information into the manuscript.

Comments 2: English Style: This manuscript contains both UK (e.g., haemoglobin) and US (e.g., hospitalized, pediatric) styles. I suggest choosing only one and making it consistent throughout the manuscript.

Response 2: We appreciate your attention to detail regarding language consistency. The manuscript has been thoroughly revised to adhere strictly to American English spelling throughout. All instances of British English spelling have been replaced with their American counterparts.

Comments 3: Influenza C: Did the test for influenza A (IAV) and B (IBV) viruses (AmpliSens Influenza virus A/B-FL kit) cover or cross-react with Influenza C?
Although this virus is rare and less symptomatic than influenza A and B, it would be interesting if your test included it.

Response 3: The AmpliSens Influenza virus A/B-FL kit used in our study is specifically designed for the detection and differentiation of Influenza A and B viruses. While the manufacturer does not provide detailed information about potential cross-reactivity with Influenza C, the test's specificity for types A and B suggests that Influenza C detection is unlikely. We acknowledge this limitation in our viral detection panel. It would be valuble to include specific tests for Influenza C to provide a more comprehensive picture of influenza infections in this population.

Comments 4: Vaccination Data: This study collects data from hospitalised paediatric patients. It would be valuable to show the vaccination data (e.g., COVID-19, influenza, Hib). This data may reveal the pattern of infection and/or coinfections in these participants.

Response 4: We agree that vaccination status is a crucial factor so we collected the data. Our study population's vaccination status was as follows:

- COVID-19: No participants under 12 years old were vaccinated, as no approved vaccine was available for this age group in Russia during the study period. For the 12-17 age group, the "Gam-COVID-Vac-M" vaccine became available in early 2022, but none of our participants had received it.

- Influenza: Approximately 2% of our sample had received influenza vaccination. All of these vaccinated individuals were infected with SARS-CoV-2, and their clinical presentations did not differ significantly from unvaccinated participants.

- Hib: Vaccination against Haemophilus influenzae type b was only introduced to the Russian national immunization schedule in 2022, so it's unlikely that study participants had received this vaccine.

Thus, due to the non-significant number of children vaccinated against these pathogens and the lack of differences between samples, we decided not to include these data in our article.

Comments 5: Table 5: Clinical Manifestations: Did this study collect data on anosmia and ageusia?
These two symptoms are prominent and seem specific to COVID-19 even in vaccinated individuals.

Response 5: We did collect data on anosmia and ageusia, recognizing their significance in COVID-19 diagnosis. However, the prevalence of these symptoms was markedly low in our cohort, with anosmia reported in only 0.5% of cases and no reported cases of ageusia. This aligns with other studies showing reduced prevalence of these symptoms in Omicron variant infections compared to earlier variants. Additionally, the young age of many participants (a large proportion under three years old) made it challenging to reliably assess these symptoms, as young children may not be able to articulate such experiences.

Comments 6: Table 5: Age Groups: I suggest subgrouping by age following Tables 2 and 4 to make it more informative.
Moreover, each age group could have different risk behaviours and lifestyles, such as infants-toddlers, preschoolers, and school-age children.

Response 6: We appreciate your suggestion to stratify the data in Table 5 by age groups. We agree that this would provide valuable insights into age-specific patterns of infection and clinical presentation. However, our current sample size limits our ability to perform this stratification while maintaining statistical significance across all subgroups.

Comments 7: Lines 68-71: I suggest adding a statement to specify the prominent waves of Omicron subvariants (e.g., BA.2.75, BA.5, etc.) during the study period. Omicron has many subvariants, and each may have different characteristic patterns.

Response 7: We agree that specifying the prominent Omicron subvariants during our study period would provide valuable context. Based on genomic surveillance data from Russia during our study timeframe (January 9, 2022, to March 10, 2023), we can add the following information: "During this period, Russia experienced distinct waves dominated by different Omicron subvariants. The initial wave (January-February 2022) was primarily driven by the BA.1 subvariant, followed by a BA.2-dominated wave in spring 2022. Subsequent waves of BA.4 and BA.5 occurred in summer and fall 2022. Towards the end of our study period, in early 2023, the XBB.1.5 sublineage began to increase in prevalence." The BA.2.75 subvariant did not become dominant in Russia. We will incorporate this information into the manuscript to provide a more comprehensive picture of the Omicron subvariants circulating during our study. We acknowledge that these subvariants may have different characteristics, which could potentially influence our findings. However, due to limitations in our testing capabilities, we were unable to differentiate between specific Omicron subvariants in our patient samples.

We appreciate your suggestion, as it will help readers better contextualize our results within the evolving landscape of SARS-CoV-2 variants.